# In Vitro Safety Study on the Use of Cold Atmospheric Plasma in the Upper Respiratory Tract

**DOI:** 10.3390/cells13171411

**Published:** 2024-08-23

**Authors:** Sigrid Karrer, Petra Unger, Michael Gruber, Lisa Gebhardt, Robert Schober, Mark Berneburg, Anja Katrin Bosserhoff, Stephanie Arndt

**Affiliations:** 1Department of Dermatology, University Medical Center Regensburg, 93053 Regensburg, Germany; sigrid.karrer@ukr.de (S.K.); petra.unger@ukr.de (P.U.); mark.berneburg@ukr.de (M.B.); 2Department of Anesthesiology, University Medical Center Regensburg, 93053 Regensburg, Germany; michael.gruber@ukr.de; 3Terraplasma Medical GmbH, 85748 Garching, Germany; lisa.gebhardt@terraplasma-medical.com (L.G.); schober@terraplasma.com (R.S.); 4Institute of Biochemistry, Friedrich-Alexander University of Erlangen-Nürnberg (FAU), 91054 Erlangen, Germany; anja.bosserhoff@fau.de; 5Comprehensive Cancer Center Alliance WERA (CCC WERA), 91054 Erlangen, Germany; 6Bavarian Cancer Research Center (BZKF), 91054 Erlangen, Germany

**Keywords:** cold atmospheric plasma (CAP), plasma intensive care (PIC), pressurized air (PA), upper respiratory tract (URT), human oral keratinocytes (hOK), human bronchial–tracheal epithelial cells (hBTE), human lung fibroblasts (hLF)

## Abstract

Cold atmospheric plasma (CAP) devices generate reactive oxygen and nitrogen species, have antimicrobial and antiviral properties, but also affect the molecular and cellular mechanisms of eukaryotic cells. The aim of this study is to investigate CAP treatment in the upper respiratory tract (URT) to reduce the incidence of ventilator-associated bacterial pneumonia (especially superinfections with multi-resistant pathogens) or viral infections (e.g., COVID-19). For this purpose, the surface-microdischarge-based plasma intensive care (PIC) device was developed by terraplasma medical GmbH. This study analyzes the safety aspects using in vitro assays and molecular characterization of human oral keratinocytes (hOK), human bronchial–tracheal epithelial cells (hBTE), and human lung fibroblasts (hLF). A 5 min CAP treatment with the PIC device at the “throat” and “subglottis” positions in the URT model did not show any significant differences from the untreated control (ctrl.) and the corresponding pressurized air (PA) treatment in terms of cell morphology, viability, apoptosis, DNA damage, and migration. However, pro-inflammatory cytokines (MCP-1, IL-6, and TNFα) were induced in hBTE and hOK cells and profibrotic molecules (collagen-I, FKBP10, and αSMA) in hLF at the mRNA level. The use of CAP in the oropharynx may make an important contribution to the recovery of intensive care patients. The results indicate that a 5 min CAP treatment in the URT with the PIC device does not cause any cell damage. The extent to which immune cell activation is induced and whether it has long-term effects on the organism need to be carefully examined in follow-up studies in vivo.

## 1. Introduction

Cold atmospheric plasma (CAP) is a partially ionized gas with bactericidal and virucidal properties [1,2,3,4,5,6]. The aim of CAP treatment is to use these properties to reduce the microbial (bacterial and viral) load in the upper respiratory tract (URT) of mechanically ventilated patients, potentially preventing superinfections of the lungs and reducing the risk of infection for nursing staff. This area of CAP application has not yet been investigated; however, this treatment offers several advantages over liquid antimicrobial agents, as the gaseous CAP can also act in hard-to-reach areas such as the subglottis.

Before clinical application, it must be assessed whether this treatment modality of CAP will cause (irreversible) tissue damage to mucous membranes or lung tissue if accidentally introduced into the lungs. Safety studies on the use of CAP in the URT are still limited. Welz et al. (2013) presented data on CAP-treated nasal and pharyngeal mucosa [7], Hasse et al. (2014) analyzed CAP application on human oral mucosa [8], and Becker et al. (2019) investigated the effect of CAP on mucosal membrane organ cultures [9]. Taken together, these studies showed that CAP treatment had no mutagenic effect. Short treatment times (up to 2 min) resulted in only a slight loss of mucosal cell viability, while the bactericidal effect was already very high. When investigating the effects of CAP on human bronchial epithelial cells, Sun et al. (2022) found a positive effect of CAP on inflammation and oxidant stress in the respiratory system, suggesting a therapeutic potential of CAP for respiratory diseases [10]. However, most studies on lung cells have been carried out on lung carcinoma cells. In this context, CAP treatment has shown clear anti-tumor potential in various studies [11,12,13]. A comparative study by Kim et al. (2016) investigating the effect of CAP on lung cancer cells (A549) and on normal human coronary artery endothelial cells (HCAEC) showed that normal and cancer cells have different sensitivities to external CAP exposure. Their results showed a CAP-dependent increase in intracellular reactive oxygen species (ROS) production in cancer cells, while normal cells were less affected, which indicates that A549 cancer cells are more responsive to CAP-mediated ROS production than HCAEC normal cells [14].

In recent years, many researchers have paid attention to the efficiency of CAP in the treatment of wounds [15,16,17,18]. There are also more and more research results describing positive CAP effects on wound healing in the oral cavity and the treatment of periodontal wounds [19,20,21,22]. This makes it all the more important to investigate the safety of CAP in the URT.

Ventilator-associated pneumonia (VAP) is one of the most frequent hospital-acquired infections occurring in intubated patients [23]. Because VAP is associated with increased mortality, morbidity, and costs, further research is required to identify effective prevention strategies. CAP is, therefore, a promising new therapeutic approach. However, this is only true if CAP can be used in the URT without causing irreversible tissue damage to the lungs or developing a toxic effect if accidentally introduced into the lungs. No data are currently available for the use of CAP in the URT. This project aims to provide initial answers to these questions.

## 2. Materials and Methods

### 2.1. Cell Lines and Cell Culture Conditions

For the present study, normal human oral keratinocytes (hOK (Gingiva); #FC-0094; CellSystems^®^ GmbH, Troisdorf, Germany), normal primary bronchial–tracheal epithelial cells (hBTE; ATTC^®^ PCS-300-010™), and normal human lung fibroblasts (hLF; #FC-0049; Cell Systems GmbH, Troisdorf, Germany) were used. hOK cells were cultured in DermaLife^®^ K Medium Complete Kit (#LL-0007; CellSystems^®^ GmbH, Troisdorf, Germany), hBTE cells were kept in Airway Epithelial Cell Basal Medium (ATCC^®^ PCS-300-030™) supplemented with Bronchial Epithelial Cell Growth Kit (ATCC^®^ PCS-300-040™); and hLF cells were maintained in FibroLife S2 Medium Complete Kit (#FC-0011; CellSystems^®^ GmbH, Troisdorf, Germany). All cells were cultured in 75 cm^2^ Falcon^®^ Cell Culture Flasks (Corning Life Sciences, Amsterdam, The Netherlands), incubated under humid conditions in a 5% CO_2_ incubator at 37 °C, and split 1:3 every 3 days. After a washing step with Dulbecco’s Phosphate Buffered Saline (DPBS) (ThermoFisher Scientific, Langenselbold, Germany), a ready-to-use solution of 0.05% trypsin/0.02% EDTA (PAN-Biotech, Aidenbach, Germany) was applied to detach the cells. The reaction was stopped with a ready-to-use trypsin inhibitor solution (PAN-Biotech, Aidenbach, Germany). After centrifugation and removal of the solution, cells were counted using Luna FL Acridine Orange/Propidium Iodide Stain (Bio-Cat GmbH, Heidelberg, Germany). Mycoplasma contamination was regularly excluded according to the manufacturer’s instructions for the PCR Mycoplasma Test Kit (PanReac AppliChem, Darmstadt, Germany).

### 2.2. Plasma Intensive Care Device and the Upper Respiratory Tract Model I

The actual application of CAP in the upper respiratory tract (URT) requires modifications (for example, the application of an air stream), which necessitate a renewed and thorough investigation of the effect of CAP on the tissues in the URT.

The respiratory tract above the cuff of an endotracheal tube has a volume of approximately 100 cm^3^ and discrete areas (i.e., oral cavity, nasal cavity, and throat) in which the CAP must be homogeneously distributed to achieve a uniform antimicrobial effect.

The developed device—the plasma intensive care (PIC)—is shown in Figure 1a and was established for a homogeneous distribution of CAP in the URT. For initial applications, a URT model I prototype consisting of four ozone-measurement points was established—nose, mouth, throat, and subglottis—to determine the ozone distribution within the system (Figure 1b). Ozone is one of the components produced by CAP and serves as an indicator of local CAP concentration.

### 2.3. Treatment of Bacteria and Cells with the Plasma Intensive Care Device in the Upper Respiratory Tract Model II

The plasma intensive care (PIC) device (Figure 2a) was used as a CAP source for the in vitro cell culture experiments and consisted of the following parts: (1) an attachment (rapid prototyping with a 3D printer) with two tube connections; (2) a 1 m long medical silicone tube going from the plasma care^®^ attachment into the URT model II; (3) a medical silicone tube going from a medical pressurized air connection to the plasma care^®^ attachment; and (4) a flow regulator that can regulate the gas flow rate of the medical pressurized air (PA) down to 0.1–1 standard liters per minute (slm).

A special URT model II (Figure 2b) was developed for bacteria and cell culture studies, in which a cell culture dish (35 mm Petri dish; Corning Life Sciences, Wiesbaden, Germany) can be placed at the “throat” and “subglottis” measurement points. The CAP-pressurized air (PA) flow, mixed via an air supply tube (Figure 2c), can be simulated realistically at these positions. The URT model II was modeled in size and volume after an adult human.

If not explicitly stated otherwise, the following cells were seeded in 35 mm Petri dishes supplemented with 2.5 mL cell culture medium under humid conditions: 150,000 hLF, 50,000 hOK, and 50,000 hBTE cells for CAP-pressurized air (PA; 0.5 slm) treatment (=PIC treatment), for pressurized air (PA; 0.5 slm) treatment, and for untreated control (ctrl.). Immediately before treatment, the cell culture medium was removed from the Petri dishes and replaced with 700 µL DPBS. This step is necessary to avoid drying effects on the cells during treatment and reflects mucous membranes moistened with saliva or tracheal fluid.

The URT model II was first opened, covered with plaster, rinsed with 20 mL DPBS, and then closed again. The opening and closing of the URT model II work magnetically. Wetting the model mimics the humid environment in the URT. The Petri dishes with the cells and the 700 µL DPBS were then clipped into the closed URT model II at the “throat” and “subglottis” measuring points.

After treatment with PA or PIC for 5 min at the measurement points “throat” and “subglottis” in the URT model II (Figure 2d), the DPBS was replaced again with 2.5 mL of the corresponding cell culture medium. Cells were cultivated for a further 24 h, 48 h, 72 h, or 96 h under humid conditions for subsequent analysis.

### 2.4. Quantification of Bacteria Inactivation after Plasma Intensive Care Treatment

A total of 100 µL of bacteria suspension (~10^6^/mL; *E. mundtii* DSM 4838; DSMZ GmbH, Braunschweig, Germany) was applied on ⌀ 9 cm Müller–Hinton agar plates (Oxoid Deutschland GmbH, Wesel, Germany) and allowed to dry for 30 min. From Müller–Hinton agar plates 3 × 35 mm (⌀) samples were punched out (per experiment), transferred to a 35 mm petri dish (Corning Life Sciences, Wiesbaden, Germany), and treated for 3 min and 6 min with PA or with the PIC device at the positions “throat” and “subglottis” in the URT model II. After subsequent incubation of the petri dishes at 37 °C for 24 h, colony-forming units (CFUs) were evaluated. For the computation of log10 reduction rates, CFUs of serial dilutions of the original bacterial suspensions at OD = 0.6 were evaluated. Each experiment was carried out at n = 3, and the results were averaged.

### 2.5. Assessment of H_2_O_2_ and NO_2_^−^/NO_3_^−^ in DPBS

For quantification of H_2_O_2_, a Fluorimetric Hydrogen Peroxide Assay Kit (Sigma Aldrich GmbH, Steinheim, Germany) was used, and the fluorescence was measured at an excitation wavelength of 540 nm and an emission wavelength of 590 nm. NO_2_^−^ and NO_3_^−^ concentrations were determined using the colorimetric Nitrite/Nitrate Assay Kit (Sigma Aldrich GmbH, Steinheim, Germany) to detect nitric oxide metabolites at 540 nm absorbance. The fluorescence was measured with a plate reader (Varioscan Flash, Thermo Fisher, Schwerte, Germany), and the Assay Kits were used as specified by the manufacturer. Each experiment was conducted n = 3, and the results were averaged.

### 2.6. Morphology of the Cells

The morphology of the hLF, hBTE, and hOK cells was assessed 24 h after a 5 min PIC treatment in the URT model II in comparison to the morphology of the untreated control (ctrl.) and the pressurized air (PA) treatment. Each experiment was carried out at n = 3, and the results were averaged. Representative images were collected with bright field microscopy (Zeiss, Axiovision, Halbergmoos, Germany) using a 20-fold magnification.

### 2.7. Viability of the Cells

To assess cell viability and exclude non-viable cells, the cells (ctrl.; PA; PIC) were stained with Acridine Orange (AO)/Propidium Iodide (PI) 24 h after treatment and analyzed using LUNA-FL™ in an automated fluorescence cell counting mode according to the manufacturer’s instructions (Logos Biosystems, Villeneuve d’Ascq, France).

### 2.8. Immunofluorescence Analysis

For immunofluorescence analysis, 5000–7500 cells were grown on Falcon^®^ 4-well culture slides (Corning GmbH, Kaiserslautern, Germany) in 1 mL cell culture medium for 24 h. Afterwards, the medium was removed, the cells were washed with DPBS, and each well was filled with 250 µL DPBS for ctrl., PA-, or PIC treatment. For this purpose, the culture slides were hermetically sealed at the “throat” and “subglottis” positions. After the treatment, DPBS was replaced with 1 mL culture medium again, and the cells were further incubated for 24 h under cell culture conditions as described in Section 2.1. To induce DNA double-strand breaks (DSBs) as a positive control, the cells were treated with etoposide (100 µM) for 3 h.

For F-actin/H2AX co-staining, the medium was tipped out of the culture slides, the chambers were removed, and the cells were washed, fixed, permeabilized, and blocked, as described previously [24]. Subsequently, the cells were incubated with Phospho-Histone H2A.X (Ser139) (20E3) Rabbit mAb (1:200; Cell Signaling, Leiden, The Netherlands) overnight at 4 °C. After washing, the cells on coverslips were incubated with the secondary antibody (1:40; fluorescein isothiocyanate (FITC)-conjugated anti-rabbit immunoglobulin; Agilent Dako, Waldbronn, Germany) for 1 h, followed by rinsing with DPBS and co-staining with 80 nmol/L rhodamine phalloidin (F-actin) (Cytoskeleton, Denver, CO, USA) in DPBS containing 10% goat serum for 40 min. After a washing step with DPBS, the cells were mounted with Vectashield Hard Set Mounting Medium with DAPI H-1500 (Vector Laboratories, Burlingame, CA, USA). Images were captured with immunofluorescence microscopy using an Axio Imager Zeiss Z1 fluorescence microscope (Axiovision Rel. 4.6.3; Carl Zeiss AG, Oberkochen, Germany) with 40-fold magnification. The experiments were repeated three times.

### 2.9. Measurement of Cell Apoptosis and Necrosis

For analysis of apoptosis and necrosis, the cells were treated with the PIC device, PA, or remained untreated (ctrl.) as described in Section 2.3. Apoptotic and necrotic cells were investigated by flow cytometry at 48 h, 72 h, and 96 h after treatment using the FITC Annexin V Apoptosis Detection Kit with PI (BioLegend, Koblenz, Germany) according to the manufacturer’s instructions. Flow cytometry analysis was conducted with a FACS Calibur Flow Cytometer (Becton Dickinson, Heidelberg, Germany). FACS data were analyzed using the FlowJo™ v10 software. The experiments were conducted in duplicate and repeated three times.

### 2.10. Migration/Proliferation Analysis

The migratory/proliferative behavior of the cells was analyzed by means of a wound healing assay using a culture insert (ibidi GmbH, Martinsried, Germany), which produces a well-defined cell-free gap between the high-density seeded (20,000 cells) of hLF, hBTE, and hOK cells in 70 µL DMEM/chamber of the culture insert. After 24 h at 37 °C and 5% CO_2_, the culture insert was removed with sterile tweezers, leaving a cell-free gap (“defined wound”) of approximately 500 µm. The dish was then covered again with a cell culture medium. The migration/proliferation into this “wound area” was documented and measured using a Carl Zeiss microscope (Carl Zeiss Vision GmbH, Halbergmoos, Germany). To determine the confluency, three areas of the defined gap from each treatment approach were photographed at 0 h, 6 h, 12 h, and 24 h. The higher the confluency, the faster the cells migrate/proliferate. The confluence of untreated cells (ctrl.) was set to 100% and was compared to PA- and PIC-treated cells for 5 min. Each analysis was performed in triplicate and repeated three times.

### 2.11. Isolation of Ribonucleic Acid (RNA) and Reverse Transcription

RNA was isolated from untreated cells (ctrl.) and from cells after PA- or PIC treatment at 24 h using the NucleoSpin RNA Plus Kit (Macherey-Nagel, Düren, Germany) according to the manufacturer’s instructions. An average value of 2–5 µg of RNA was then transcribed into cDNA by reverse transcriptase reaction using the Super ScriptTM II Kit (Invitrogen, Thermo Fisher Scientific, Emeryville, CA, USA).

### 2.12. Quantitative Real-Time Polymerase Chain Reaction (PCR) Analysis

Gene expression analysis consisting of quantitative real-time PCR with specific primer sets (Sigma-Aldrich, Steinheim, Germany) and conditions (Table 1) was performed using LightCycler technology (Roche Diagnostics, Mannheim, Germany) as described elsewhere [25]. The PCR reactions were evaluated by melting curve analysis. Beta-actin (β-actin) was amplified to ensure cDNA integrity and to normalize expression. Each experiment was repeated at least three times in duplicate.

### 2.13. Statistical Analysis

All data were analyzed with GraphPad Prism 5 software (GraphPad Software Inc., San Diego, CA, USA) and expressed as mean ± standard deviation (SD). Ordinary one-way ANOVA with Tukey’s multiple comparison tests was performed to indicate differences in the mean within the ctrl., PA-, and PIC-treated groups at both positions (“throat” and “subglottis”). Significant results are indicated by * *p* ≤ 0.05, ** *p* < 0.01, *** *p* < 0.001, or **** *p* < 0.0001.

## 3. Results

To evaluate the safety of PIC in the URT, special URT models were developed, which correspond in size and volume to an adult human. The URT model I was used to determine the ozone concentration at different measuring points. The URT model II was used to test the antibacterial efficacy and to perform various cell culture experiments with hLF, hBTE, and hOK cells.

### 3.1. Ozone Measurements in the URT Model I

Initial studies with the PIC device (Figure 1a) and the URT model I (Figure 1b) have shown that with an airflow of 0.5 slm, a homogeneous ozone concentration of 200–400 parts per million (ppm) is achieved at all measurement points within 90 s (Figure 3). The maximum ozone concentration achieved at the “subglottis” measurement point (approximately 250 ppm) corresponded to approximately half of the ozone concentration achieved at the “mouth” measurement point (approximately 400–450 ppm). It was also shown that the atmosphere in the URT model I was cleared again within a very short time after the end of CAP production (CAP off) (Figure 3).

### 3.2. Antibacterial Efficacy of the PIC Device

The bactericidal effect of the PIC device was investigated at the “throat” and “subglottis” measuring points using *Enterococcus mundtii* (*E. mundtii*) as an example. At both measuring points, a germ reduction of at least 3–4 log (=99.9 to 99.99%) was achieved after just 3 min of plasma application (Figure 4), which means the following: (1) Flooding the URT with CAP can significantly reduce the microbial load and thus potentially prevent nosocomial pneumonia, and (2) not only ozone but also other CAP species play an important role in the antimicrobial effectiveness of PIC treatment because almost similar log reductions in bacteria were achieved at both measuring points (“throat” and “subglottis”), although the ozone concentration at the “subglottis” measuring point was significantly lower (Figure 3).

### 3.3. Reactive Species-Measurements in the URT Model II

ROS and RNS are described as decisive components of CAP. To examine which RONS are produced in DPBS after treatments (ctrl., PA, PIC) at the positions “throat” and “subglottis” in the URT model II, long-lived ROS were determined based on scaling H_2_O_2_ and RNS by measuring nitrite (NO_2_^−^) and nitrate (NO_3_^−^). These species were analyzed in DPBS since all cell treatments in the URT model II were carried out in DPBS to avoid drying effects during treatments. Measurements showed stimulation effects of H_2_O_2_ (Figure 5a), NO_2_^−^ (Figure 5b), and NO_3_^−^ (Figure 5c) after PIC treatment with much higher inductions at the “throat” position compared to the “subglottis” position. These results are consistent with the ozone measurements, which also show higher values at the “throat” position (Figure 3).

### 3.4. Cell Morphology Is Not Changed after PIC Treatment

Before starting the risk assessment of the PIC device on human cells in vitro, several preliminary experiments were performed to establish the experimental design and the implementation of the cell treatment in the URT model II. First, it was investigated whether a 5 min treatment with pressurized air (PA; 0.5 slm) had any effect on the cells. Morphological examinations were performed immediately (0 h) and 24 h after treatment and showed no changes in each of the three cell types. A PIC treatment of up to 5 min also showed no morphological changes in each of the three cell lines analyzed compared to the untreated control (ctrl.) (Figure 6).

### 3.5. Cell Viability Is Not Influenced after PIC Treatment

The total cell count and the number of live and dead cells were determined at both positions (“throat” and “subglottis”) after a 5 min treatment with PA or the PIC device and were compared to the untreated control (ctrl.). To avoid a loss of the total cell count after CAP treatment due to a temporary loss of attachment, which has already been described previously by various working groups [26,27,28,29], the medium change was only carried out after the re-attachment of the cells (approximately 20 min after treatment). As a result, no increase in dead cells was detected in the “throat” and “subglottis” positions 24 h after PA- or PIC treatment in hLF (Figure 7a) or hBTE (Figure 7b) and hOK cells (Figure 7c) in comparison to the untreated control (ctrl.), which means that viability is essentially unaffected by any of the treatments.

### 3.6. DNA Is Not Damaged after PIC Treatment

To determine DNA damage after PA- or PIC treatment, hBTE and hOK cells and hLF were double stained with H2AX/F-actin 24 h after treatment. H2AX is a marker for DNA damage, and F-actin stains the cytoskeleton of cells. The cancer chemotherapeutic drug etoposide, which induces DNA double-strand breaks (DSBs), was used as a positive control. The damaged cell nuclei show an intense green color. A 5 min treatment with PA or the PIC device does not result in DNA damage or morphological changes in hLF (Figure 8a) or hOK cells (Figure 8b). hBTE cells start to differentiate at passage 6. At lower passages, the cells do not adhere to the chamber slides for immunofluorescence staining. Therefore, no DNA damage could be determined for this cell type using the H2AX immunofluorescence staining method.

### 3.7. Cell Apoptosis Is Not Affected after PIC Treatment

Using FACS PI/Annexin V analysis, apoptosis and necrosis of the cells were determined 48 h after 5 min of PA- or PIC treatment and in untreated cells. Necrosis was virtually undetectable in all treatment groups and cell types (Figure 9a–c). It should be noted that hBTE (Figure 9b) and especially hOK cells (Figure 9c) showed increased levels of basal apoptosis compared to hLF (Figure 9a), even without PIC treatment. However, no further significant induction of apoptosis was observed after 5 min of PIC treatment in any of the cell lines analyzed. To exclude potential long-term damage effects due to PIC treatment, the experiments were repeated with an extended incubation time (72 h and 96 h) after treatments. However, no changes in apoptosis and necrosis values were observed after 72 h and 96 h in prior experiments, suggesting that PIC treatment does not cause any long-term damage to the analyzed cells.

### 3.8. Cell Migration/Proliferation Is Not Significantly Modified after PIC Treatment

A wound healing assay was used to investigate the migration/proliferation of hLF, hBTE, and hOK cells after 5 min of PA- and PIC treatment. To determine the confluency, three areas of the “scratch” were photographed 0 h, 6 h, 12 h, and 24 h after treatment. The confluency of the cells was determined at two defined locations per image. The higher the confluency, the faster the cells’ ability to migrate/proliferate. As soon as the “scratch” is no longer visible (in this case, after 24 h), the confluency is 100%. A 5 min treatment with the PIC device does not lead to any significant changes in the migration ability of hLF (Figure 10a,b) and hOK cells (Figure 10c,d). However, a slight but not significant increase in migration was observed in the untreated hLF control cells between 6 h and 12 h (Figure 10b). The migration/proliferation behavior of hBTE could not be determined in the wound healing assay because the cells detached from the surface when the cell density was too high.

### 3.9. Immunomodulatory Effects Are Recognized after PIC Treatment

It is known that CAP can influence the release of inflammatory cytokines, chemokines, and other immunomodulatory molecules, which may lead to the recruitment of immune cells [15,30,31,32]. As a result, a local increase in immune cells can potentially have immunomodulatory effects and, for example, intensify cytokine release syndrome or macrophage stimulation syndrome (CRS/MSS). However, this is not desirable in the context of using the PIC device in the URT. Studies have also shown that CAP can influence the expression of fibrosis-associated molecules in fibroblasts [33,34]. To investigate the effect of PIC treatment on inflammation and fibrosis, the gene expression profiles of pro-inflammatory cytokines in hBTE and hOK cells and profibrotic molecules in hLF were examined at the mRNA level.

#### 3.9.1. Induction of Pro-Inflammatory Factors in hBTE and hOK Cells after PIC Treatment

Monocyte chemoattractant protein-1 (MCP-1/CCL2) is one of the key chemokines that regulate the migration and infiltration of monocytes and macrophages. Migration of monocytes from the bloodstream across the vascular endothelium is required for routine immunological surveillance of tissues as well as in response to inflammation [35]. Using mRNA expression analysis, we observed a significant increase in MCP-1 expression in hBTE cells at the “throat” and “subglottis” positions (Figure 11a) but no increase in hOKs after PIC treatment (Figure 11d).

Interleukin-6 (IL-6), a member of the pro-inflammatory cytokine family, not only induces the expression of a variety of proteins responsible for acute inflammation but also plays an important role in the proliferation and differentiation of cells in humans. Vice versa, an overproduction of IL-6 and dysregulation of the IL-6 signaling pathways can result in inflammatory and autoimmune disorders as well as in the development of cancer, suggesting that IL-6 plays an important role in the human cytokine network [36]. An increase in IL-6 expression was observed in both cell lines after PIC treatment at both positions (Figure 11b,e).

Tumor necrosis factor alpha (TNF-α) is the most widely studied pleiotropic cytokine of the TNF superfamily. Under pathophysiological conditions, the generation of high levels of TNF-α leads to the development of inflammatory responses that are hallmarks of many diseases. Of the various pulmonary diseases, TNF-α is implicated in asthma, chronic bronchitis, chronic obstructive pulmonary disease, acute lung injury, and acute respiratory distress syndrome. In addition to its underlying role in inflammatory events, there is increasing evidence for the involvement of TNF-α in cytotoxicity [37]. Here, both analyzed cell lines showed a significant increase in TNF-α at both measurement points after PIC treatment (Figure 11c,f).

#### 3.9.2. Induction of Profibrotic Molecules in hLF after PIC Treatment

Excessive deposition of extracellular matrix (ECM) in the alveolar region and increased oxidative stress (e.g., exogenous ROS sources) are two central pathological features in pulmonary fibrosis [38,39]. For example, myofibroblasts synthesize and deposit excessive amounts of ECM proteins, such as collagen type I (collagen-I) [40,41]. Although we did not observe any morphological signs of an activated myofibroblast phenotype (Figure 8a; F-actin staining), surprisingly, there was a significant increase in collagen-I expression at the mRNA level in hLF after PIC treatment in both positions (“throat” and “subglottis”) (Figure 12a).

In addition, the myofibroblast marker α-smooth muscle actin (αSMA) was significantly increased after PIC treatment (Figure 12b), which also indicates hLF activation.

FK506-binding protein 10 (FKBP10, also termed FKBP65), a member of the family of immunophilins, is an endoplasmic reticulum (ER)-resident peptidyl-prolyl isomerase and a collagen-I chaperone [42]. FKBP10 was reported to be upregulated in experimental lung fibrosis and idiopathic pulmonary fibrosis (IPF), where it is mainly expressed by (myo)fibroblasts [43]. FKBP10 in hLF was significantly induced after PIC treatment at both measurement points (Figure 12c), which also indicates fibroblast activation.

## 4. Discussion

Several studies have already confirmed that cold atmospheric plasma (CAP) has antibacterial and antiviral properties [1,2,3,4,5,6,44,45,46] and that, when used appropriately, it does not induce mutagenic or toxic effects on normal eukaryotic cells [47]. In addition, the study by Evert et al. (2021) showed that repeated exposure of murine oral mucosa to CAP was well tolerated and did not result in carcinogenic effects, motivating future applications of CAP in patients for dental and implant treatments [48].

However, to ensure the safety of use in the upper respiratory tract (URT), certain characteristics of the lung mucosa must be taken into account in order to exclude a harmful effect in this physiological system. It must be ensured that a high level of pro-inflammatory mucosal activation is not induced and that hyper-inflammatory reactions do not occur. In addition, cell death (apoptosis or necrosis) induced by CAP treatment in the URT must be avoided. Furthermore, morphological changes in the epithelial structure of the cell layer and mutations, for example, in the sense of DNA double-strand breaks (DSBs), must be avoided to prevent fibrotic remodeling of the lung tissue. Furthermore, it is necessary to investigate how CAP spreads in the respiratory tract system and what dose and frequency of CAP should be used. All these questions must be answered before CAP can be used clinically in the URT of patients.

This preclinical in vitro study on hOK and hBTE cells and hLF is the first study to assess the use of CAP in the URT and to provide initial information on the safety of CAP use in this system.

Flooding the URT with the plasma intensive care (PIC) device (CAP + pressurized air (PA)) for 5 min at the “throat” and “subglottis” positions (Figure 2) showed no significant differences to the untreated control with regard to cell morphology, viability, DNA damage, apoptosis, and migration (Figure 6, Figure 7, Figure 8, Figure 9 and Figure 10). These results indicate that a single 5 min PIC treatment does not trigger any cellular effects in vitro. Long-term studies (72 h and 96 h after PIC treatment) also showed no increased apoptosis or necrosis as a result of PIC treatment. For this reason, the setting for the safe CAP application has been programmed to 5 min. However, in establishing studies to find the appropriate CAP dose, we observed slight signs of cell-damaging effects (apoptosis, necrosis, and DSBs) after a 10 min PIC treatment in prior experiments; for this reason, we would not recommend treatment times of more than 5 min, especially since the antibacterial effect of CAP already achieves very good results at much shorter treatment times, as described in different studies [7,8,9] and also observed in the present work (Figure 4). When comparing untreated controls with PA-treated cells, no cellular differences were seen either. These results suggest that PA (0.5 standard liters per min (slm)) does not have any cellular effect on the analyzed cell types. Because 0.5 slm is sufficient to distribute the CAP evenly in the URT within 90 s, no higher dose of pressurized air is necessary.

An important effect that should not be neglected is the drying effect during treatment with PA or PIC when using 0.5 slm pressurized air. To prevent cell damage due to drying, the cells were covered with a DPBS film during treatment, which reflects the moist environment in the URT. In addition, the interior of the model was also moistened with DPBS during treatment. Future studies using enriched fluids (e.g., sugars, lipids, proteins) or saliva could more realistically mimic the moist environment of the mucosa.

Initial studies with the PIC device and the URT model I have shown that a homogeneous ozone concentration of 200–400 ppm is achieved with an airflow of 0.5 slm within 90 s (Figure 3). This ozone concentration is approximately half of that used in superficial wound treatment. The maximum ozone concentration at the “subglottis” measurement point was approximately 250 ppm, which is about half of the maximum ozone concentration achieved at the “throat” measurement point (approximately 450 ppm). These results suggest that the concentration of reactive species in the URT model decreases with increasing distance from the oral cavity. These results were confirmed by measuring reactive oxygen and nitrogen species (RONS) (Figure 5). Since we did not observe any cellular differences between the two measurement points, we assume that the production of RONS throughout the URT model within a 5 min treatment is in the safe range.

The investigations into the antibacterial effect of PIC in the URT model II also observed a sufficient reduction in Colony-Forming Units (CFUs) (reduction of 99.9–99.99%) after just 3 min of PIC treatment. Slightly better effects were observed at the “throat” position. This can be explained by the increased amounts of ozone and other RONS at this position compared to the “subglottis” position.

Interestingly, the mRNA expression profile of various pro-inflammatory (MCP-1, IL-6, TNFα) and profibrotic (collagen-I, FKBP10, and αSMA) genes was significantly changed after PIC treatment (Figure 11 and Figure 12). The induction of these genes appears to be CAP-regulated, but we did not detect any relevant differences between the “throat” and “subglottis” measurement points, which suggests that the dose of RONS at both measurement points is sufficient to initiate these changes at the molecular level.

It is known from other studies that CAP can influence these groups of genes and lead to the local release of different cytokines, chemokines, and growth factors [30,49,50]. Follow-up studies should clarify whether PIC treatment also induces strong activation of these proteins in vivo and whether such activation may even cause long-term changes (e.g., fibrosis).

In the case of COVID-19 patients with a critical disease course, it must be taken into account that some of them show signs of cytokine release syndrome (CRS) or macrophage activation syndrome (MAS) [51,52]. A secondary effect of CAP treatment is the release of pro-inflammatory cytokines, which could exacerbate CRS and MAS. In addition, it should be considered that in clinical applications, repeated CAP treatments may be performed, which could further exacerbate immune cell activation. This potential risk should be considered and excluded before the clinical use of CAP in the URT can be approved.

## 5. Conclusions

CAP is a rapidly growing new research area in health care. One promising new medical application of CAP is its use in the URT to reduce the incidence of ventilator-associated bacterial pneumonia and viral infections.

This in vitro safety assessment study provides the first investigation into the use of CAP in this area. The results indicate that CAP treatment in the URT with the PIC device was successful under established conditions without any signs of cell damage. The extent to which immune cell activation is induced and whether it has long-term effects on the organism need to be carefully examined in follow-up studies.

As soon as it has been demonstrated in vivo that the use of CAP in the URT is safe and effective, it can make an important contribution to the recovery of intensive care patients and represent a further component of hospital hygiene.

## 6. Patents

WO2022008684A1 (System and plasma for treating and/or preventing a viral, bacterial, and/or fungal infection).

## Figures and Tables

**Figure 1 cells-13-01411-f001:**
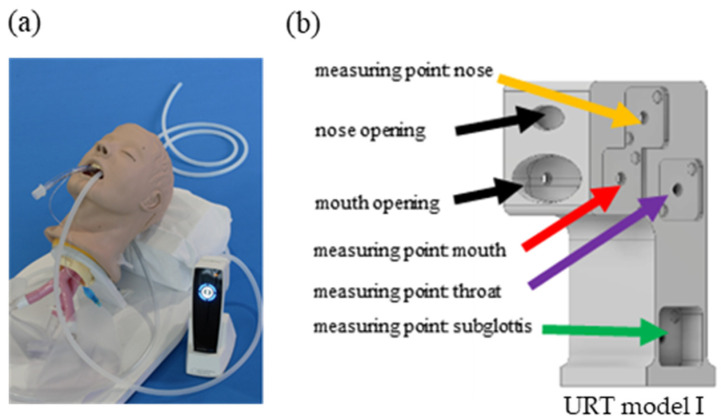
The plasma intensive care (PIC) device and the upper respiratory tract (URT) model I. (**a**) The PIC device; (**b**) URT model I with four ozone-measurement points (nose, mouth, throat, and subglottis) and two additional openings (nose and mouth).

**Figure 2 cells-13-01411-f002:**
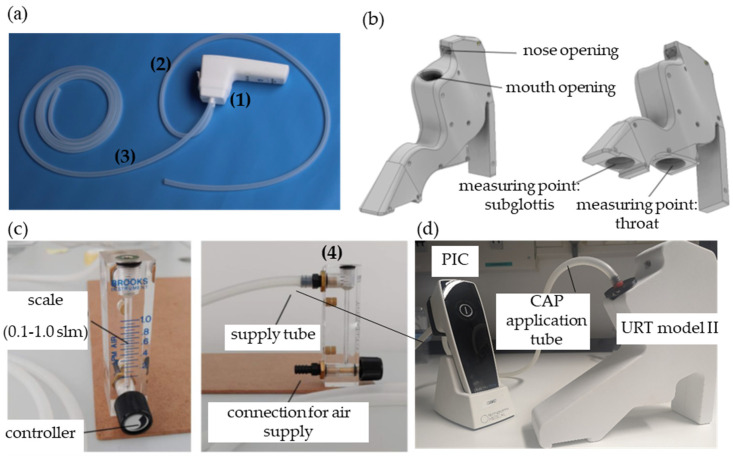
Plasma intensive care (PIC) device and the upper respiratory tract (URT) model II for bacteria and cell culture treatments. (**a**) The PIC device with an attachment **(1)** for two silicone tubes. The application tube **(2)** (short tube on the right) is connected on one side, and the supply tube **(3)** (long tube on the left) is connected on the other side. (**b**) URT model II with mouth and nose openings to apply the CAP. The two positions (measurement point subglottis and throat) for the attachment of cell culture dishes can be seen. (**c**) Flowmeter and supply tube connection **(4)**. The flow rate can be adjusted between 0.1 and 1 standard liter per minute (slm) using the rotary control. It should be noted that the connection nozzles are designed for low pressure; therefore, the air supply must be virtually pressure-free. (**d**) Experimental setup: PIC device with air supply tube attachment and application tube in the URT model II for cell culture treatments. The application tube is positioned in the mouth opening on the URT model II with an adapter.

**Figure 3 cells-13-01411-f003:**
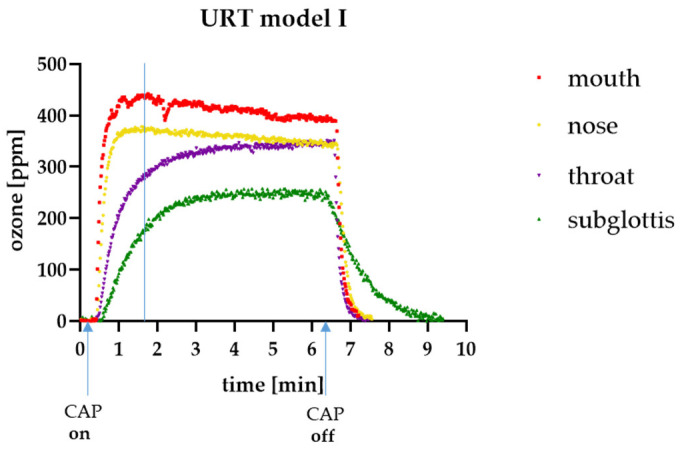
Ozone distribution in the URT model I. Measurement of the ozone concentration in parts per million (ppm) at different measurement points in the URT model I using a 1 m long silicon tube. The blue line in the table shows the time point (after about 90 s) when an ozone concentration of approx. 200–400 ppm was achieved at all measurement points.

**Figure 4 cells-13-01411-f004:**
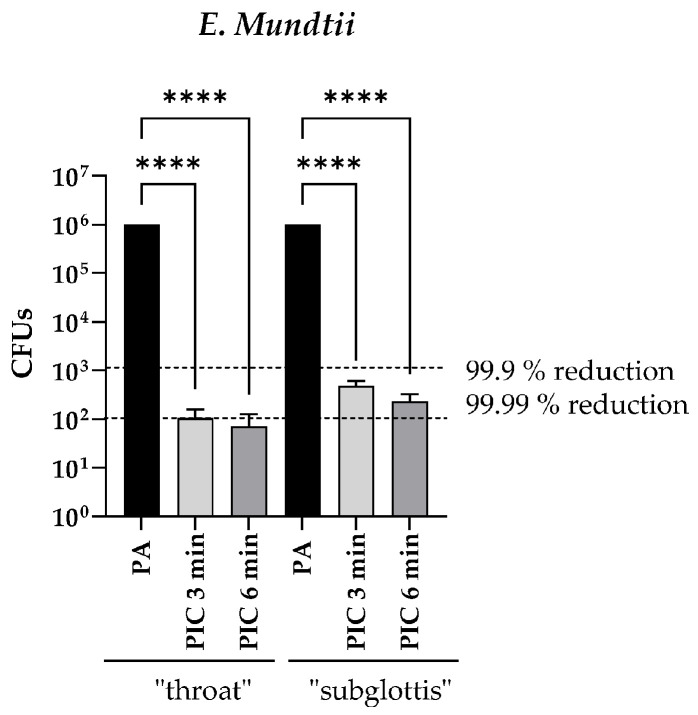
Decolonization of *E. mundtii* using the PIC device. The Colony-Forming Units (CFUs) from *E. mundtii* were determined after PA treatment (6 min; 0.5 standard liters per minute (slm)) and after 3 min and 6 min of PIC treatment at the “throat” and “subglottis” positions in the URT model II. Black dotted lines indicate the reduction in three log10 steps of viable bacteria (99.9%) and the reduction in four log10 steps of viable bacteria (99.99%). (n = 3, median ± interquartile range). **** *p* < 0.0001.

**Figure 5 cells-13-01411-f005:**
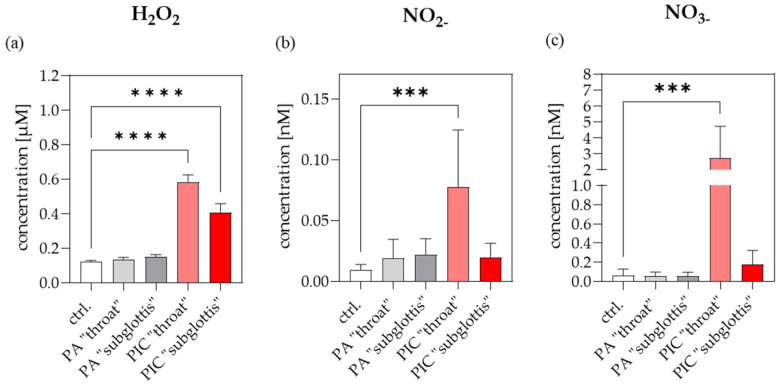
H_2_O_2_, NO_2_^−^ and NO_3_^−^ in DPBS. A total of 700 µL of DPBS was treated (ctrl., PA, PIC) at both positions (“throat” and “subglottis”) for 5 min (n = 3). (**a**) Using an H_2_O_2_ standard series, the Fluorometric Hydrogen Peroxide Assay Kit was used to determine the H_2_O_2_ concentration [µM] in DPBS. (**b**) NO_2_^−^ and (**c**) NO_3_^−^ concentrations [nM] in DPBS were quantified using a colorimetric Nitrite/Nitrate Assay Kit. Statistical analysis: Ordinary one-way ANOVA with Tukey’s multiple comparison test was performed to compare the mean of untreated ctrl. to PIC treatment at both positions. *** *p* < 0.001, **** *p* < 0.0001.

**Figure 6 cells-13-01411-f006:**
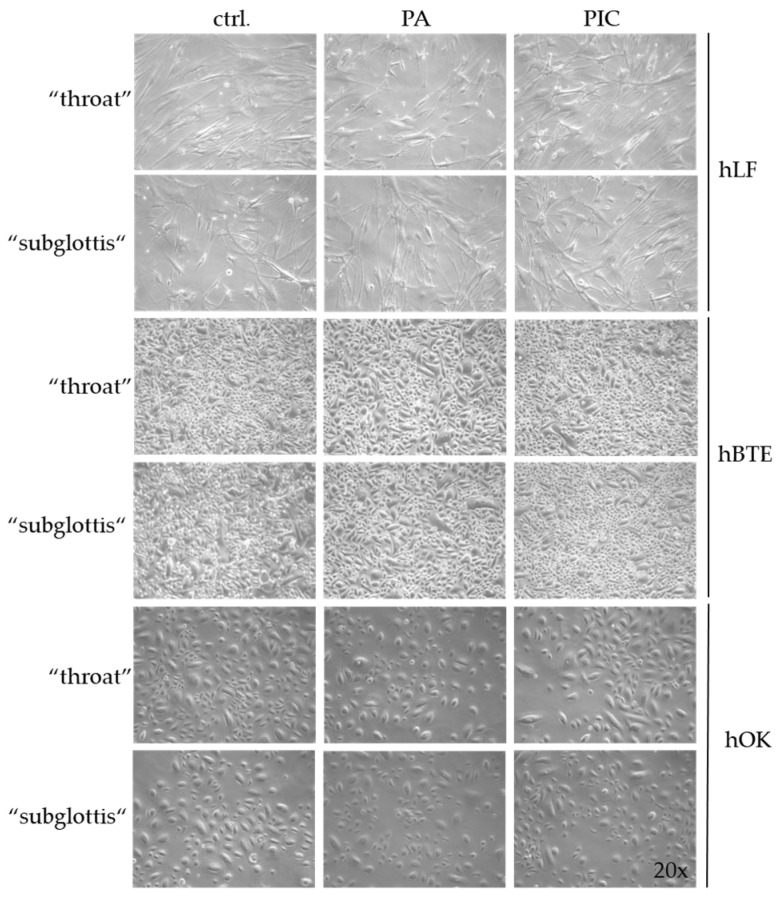
Morphology of hLF and the hBTE and hOK cells photographed 24 h after 5 min of PA treatment (0.5 standard liters per minute (slm)) or after 5 min of PIC treatment. Results were compared to the untreated control (ctrl.) at the “throat” and “subglottis” positions in the URT model II. Light microscopic images were taken at 20-fold magnification.

**Figure 7 cells-13-01411-f007:**
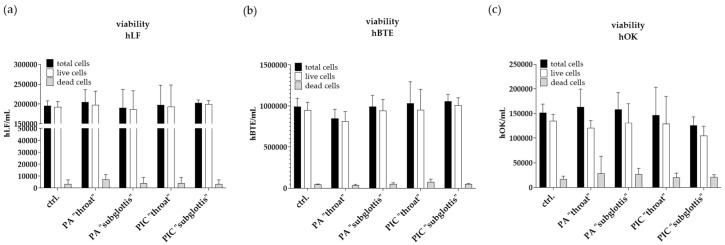
Viability determination of (**a**) hLF, (**b**) hBTE, and (**c**) hOK cells 24 h after PA- and PIC treatment in comparison to the untreated ctrl. in the URT model II at the “throat” and “subglottis” positions using Acridine Orange/Propidium Iodide staining and measurement in LUNA-FL™ in an automated fluorescence cell counting mode. The results of the three experiments were averaged.

**Figure 8 cells-13-01411-f008:**
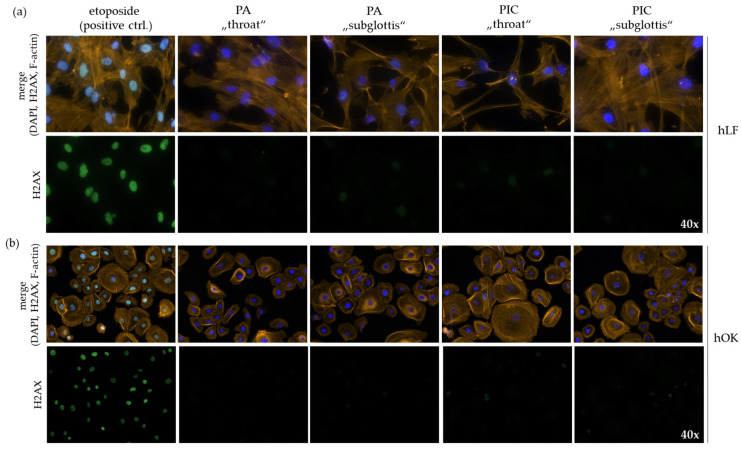
H2AX (Fitc; green) immunofluorescence staining for the detection of DNA damage in (**a**) hLF and (**b**) hOK cells after a 5 min PA- or PIC treatment at the “throat” and “subglottis” positions in the URT model II in comparison to the etoposide (100 µM) positive control. F-actin (rhodamine; orange) stains the cytoskeleton of the cells. Dapi (blue) stains the cell nuclei—40-fold magnification.

**Figure 9 cells-13-01411-f009:**
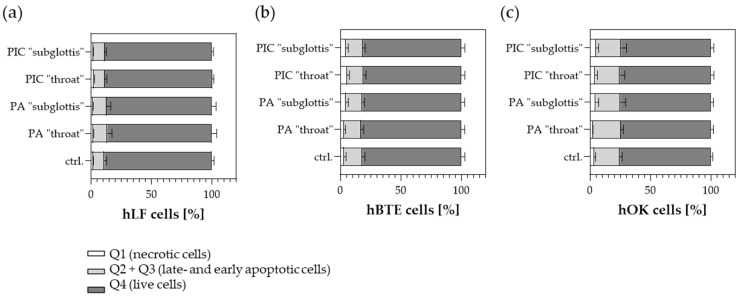
Annexin V/PI double-staining assay was performed 48 h after PA- and PIC treatment on the “throat” and “subglottis” positions in the URT model II and were compared to untreated control cells (ctrl.). Apoptosis (early and late apoptosis), necrosis, and the number of live cells were analyzed in (**a**) hLF, (**b**) hBTE, and (**c**) hOK cells. The graphs present the percentage (mean ± SD) of the cells in the region among the total cells from three independent experiments in duplicate.

**Figure 10 cells-13-01411-f010:**
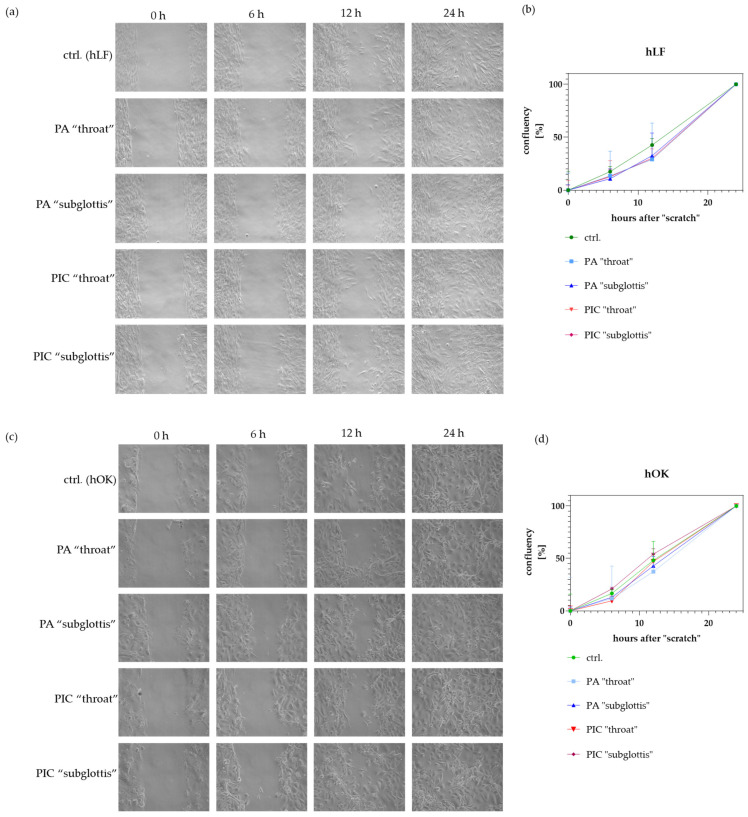
Wound healing assay with hLF and hOK cells after 5 min of PA- and PIC treatment in comparison to untreated control (ctrl.) cells. Wound closure was monitored over 24 h. Representative images of the wound healing progress of (**a**) hLF and (**c**) hOK cells are shown over time (0 h, 6 h, 12 h, and 24 h). The graphs summarize the confluency (in %) of three independent experiments performed with (**b**) hLF and (**d**) hOK cells at the “throat” and “subglottis” positions. Light microscopic images were taken at 20-fold magnification.

**Figure 11 cells-13-01411-f011:**
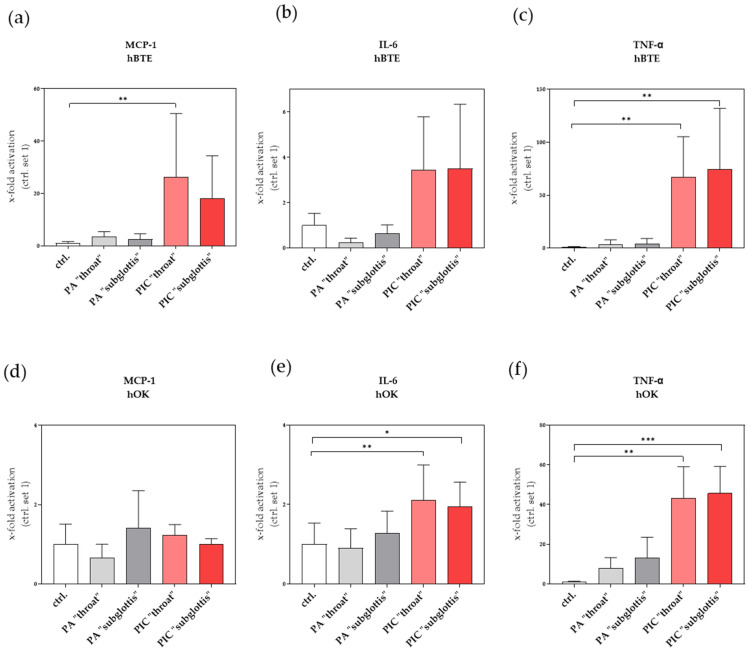
Effects of PIC treatment on pro-inflammatory gene expression in hBTE (**a**–**c**) and hOK (**d**–**f**) cells. The relative mRNA expression of MCP-1 (**a**,**d**), IL-6 (**b**,**e**), and TNF-α (**c**,**f**) was analyzed 24 h after PA- and PIC treatment. Statistical analysis: An ordinary one-way ANOVA with Tukey’s multiple comparison test was performed to compare the mean of untreated ctrl. to PA- and PIC treatments at both positions. * *p* ≤ 0.05, ** *p* < 0.01, *** *p* < 0.001.

**Figure 12 cells-13-01411-f012:**
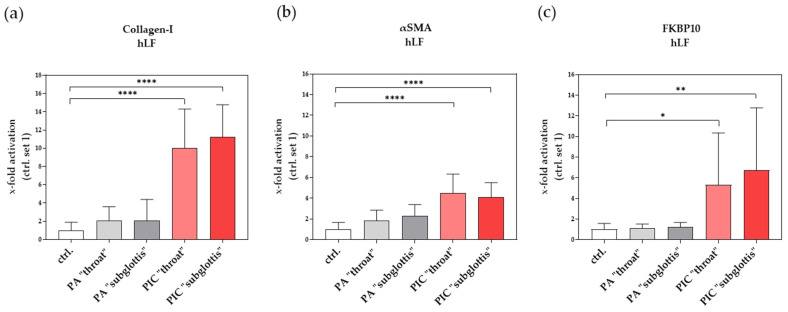
Effects of PIC treatment on profibrotic gene expression in hLFs. The relative mRNA expression of collagen-I (**a**), αSMA (**b**), and FKBP10 (**c**) was analyzed 24 h after PA- and PIC treatment. Statistical analysis: An ordinary one-way ANOVA with Tukey’s multiple comparison test was performed to compare the mean of untreated ctrl. to the PA- and PIC treatments at both positions. * *p* ≤ 0.05, ** *p* < 0.01, **** *p* < 0.0001.

**Table 1 cells-13-01411-t001:** Human primers and conditions.

Primer Name	Forward Primer 5′→3′	Reverse Primer 5′→3′	Condition ^1^ (Annealing, Melting)
β-actin	CTACGTCGCCCTGGACTTCGAGC	GATGGAGCCGCCGATCCACACGG	ann. 60 °C, melt. 85 °C
Coll I	CGGCTCCTGCTCCTCTT	GGGGCAGTTCTTGGTCTC	ann. 60 °C, melt. 86 °C
αSMA	GGCCGAGATCTCACTGACTAC	TTCATGGATGCCAGCAGA	ann. 58 °C, melt. 84 °C
FKBP10	TGCGGATGTGGTGGAAATCA	CCGTAGTCATGCGAGGTGAA	ann. 60 °C, melt. 81 °C
IL-6	GGTACATCCTCGACGGCATCT	GTGCCTCTTTGCTGCTTTCAC	ann. 60 °C, melt. 79 °C
TNF-α	ATCCTGGGGGACCCAATCTA	AAAAGAAGGCACAGAGGCCA	ann. 60 °C, melt. 81 °C
MCP-1	AATCAATGCCCCAGTCACCT	GGGTCAGCACAGATCTCCTT	ann. 60 °C, melt. 82 °C

^1^ Quantitative real-time PCR was conducted with specific sets of primers and conditions. ann: annealing temperature; melt: melting temperature.

## Data Availability

Data are available upon reasonable request from the first and corresponding authors.

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
