# Peer review of "In Vitro Safety Study on the Use of Cold Atmospheric Plasma in the Upper Respiratory Tract"

_cells, 2024, doi:10.3390/cells13171411_

Round 1

Reviewer 1 Report

Comments and Suggestions for Authors

The manuscript is well written and gives precise information about the CAP device as well as cellular effects. However, it lacks some critical points.

The manuscript needs major revision, before it can be accepted:

1.   The overall message is that the current treatment regime does not cause impaired cell viability and DNA damage. However this is based on experiments with very short incubation times. DNA damage is known to cause effects only after some cell divisions, that is why e.g. colony forming assays with an incubation time of 21d are still a gold standard in Radiotherapy. Please include long-term experiments at least for Viability /Apoptosis experiments.

2.   The authors discuss that the anti-bacterial/viral efficacy of the device has been tested (Ref 7-9) and that this is an argument why 5 min treatment is enough. However it seems that the antibacterial tests have been performed with a different setup (without carrier gas; much shorter distance to the mesh electrode, dires sample). How can this 5 min treatment be related to the current study, where the distance to the electrode was much larger, the sample was coated with DPBS and pressurized air was used. Much longer treatment times might be needed to effectively eliminate bacteria and the results of the current study might not be transferrable.

3.    The Introduction contains  a detailed description of the device -line 66-81. Please move to Methods section . Give more information on CAP effects on gingiva; bronchial/tracheal; fibroblast cells or other (pre-)clinical studies.

4.    Line 132-3: The authors state that the DPBS resembles the moist environment of the mucosa. However, the composition of the mucosa is more complex. The effects of CAP may be different if e.g. proteins are in the surrounding fluid. Why did the authors not include proteins /sugar/lipids in the fluid? Please repeat part of the experiments with more “realistic” conditions and compare.

5.    The authors state in line 452-5 that RONS are “assumed to be in the safe range”. Please measure the most abundant RONS to support the statement. RONS generation largely depends on distance to nozzle and thus papers with different device geometry should not be cited.

6.    Discussion: The authors suggest in vivo studies for further experiments. please comment on the further possibilities regarding the 3R principles. What is state-of the art for efficacy and safety testing before in vivo study. Would it e.g. be possible to use Co-culture models of infected mucosa in the current device?

7.    Fig 7: part d-f does not give any new information. All data are already shown in 7a-c. Please move the plots to supplementary data or remove.

8.    Fig 8: b& d: Axis is declared as “migration”. However the gap closure may be due to migration and proliferation of cells. I would suggest to name the axis: “confluency” or “gap size”

9.    Fig 9 /10 has very high standard variation for PIC groups. Please give explanation. How did the authors ensure the “quality! Of the plasma was identical for all experiments. Did the authors perform spectroscopy or similar to ensure reproducibility of each PIC treatment?

Reviewer 2 Report

Comments and Suggestions for Authors

The submitted paper presents a preclinical in vitro study on cells of the upper respiratory tract (URT), to assess the use of cold atmospheric plasma (CAP) in the URT and to provide initial information on the safety of CAP use in this system.

The study is well organized and clearly presented; the papers is properly balanced, suitably introduced and with an exhaustive set of references. Language is good. Figures are clear, well explained and linked to the text; some fonts are too small (e.g.Fig.7 d-e-f)

Some specific comments are listed in the following:

Line 59 “ROS” is not spelled; please double check you have properly defined all non obvious acronyms

Lines 66-72 – four parts of the PIC device are listed in the text: it would be easier to read if they are also labeled in Fig.1a and Fig.2a

Line 74 “In contrast to a wound area to be treated, … five times the volume” – please better explain the term of comparison: how is this area typically defined? Is there a reference?

Line 91 “No data are currently available for the PIC device.” – The sentence raises the question: and more in general, apart from PIC device, are there other studies that have faced the issue defined in the previous sentence (… if CAP can be used in the URT without causing irreversible tissue damage to the lungs …) ?

The question is important because the final sentence “This project aims to provide initial answers to these questions.” Is a quite heavy statement.

If you consider it proper, please discuss this point in some more detail, as the level of originality of this work depends on it.

Line 134 “Before treating the cells at the “throat” and “subglottis” measurement points in the URT model II, the model was opened, lined with plasters, and rinsed …” – Please explicitly confirm if this is performed before or after positioning the cells, as it is not so clear from the text

Fig.6 – First and third rows do not seem to be discussed in the text, at least not clearly enough

Round 2

Reviewer 1 Report

Comments and Suggestions for Authors

Th authors have answered all questions and changed the manuscript according to the remarks. Additional experiments have improved the quality of the paper.